# OpenReview forum: "Don't Trust: Verify -- Grounding LLM Quantitative Reasoning with Autoformalization"
_ICLR.cc/2024/Conference — ICLR 2024 poster_

### Official Review · Reviewer_1bgq · 2023-10-28

**Soundness:** 3 good
**Presentation:** 3 good
**Contribution:** 1 poor
**Rating:** 3
**Confidence:** 4

**Summary:**

This paper combines large language models (LLMs) and theorem provers to improve the symbolic reasoning capabilities of the former. Basically, LLMs translate mathematical statements into Isabelle code, which is then used to check for correctness.
The proposed approach is evaluated experimentally with favourable results compared to the state-of-the art.

**Strengths:**

1) The proposed translation includes both symbolic and neural filters to improve the statement translation reliability.

2) The related work is discussed in detail.

**Weaknesses:**

1) The originality and theoretical contribution of this paper is rather limited, as this is mainly the clever combination of two existing tools.

2) The experimental comparison is mainly wrt vanilla voting methods. Also, there is no explanation about the reasons of the particular approached selected for comparison, and there is no comparison with the numerous other methods mentioned in the related work.

**Questions:**

p. 3: Is it the case that every yes/no question is translated into a request for proof?
What if I ask "Is there a proof that \pi is irrational?"

---

> ### Author Response · Authors · 2023-11-15
> **Response to Reviewer 1bgq**
>
> Thank you for your constructive feedback and we address individual points below.
>
> > The originality and theoretical contribution of this paper is rather limited, as this is mainly the clever combination of two existing tools.
>
> Thank you for your thoughts and we would like to provide some clarifications. Although the core components of DTV are indeed large language models and theorem provers in the formal environment, we disagree with the conclusion that the contribution of the paper is limited. Several other reviewers have expressed and liked the originality and simplicity of our approach. We also note that using theorem provers as a tool allows us to solve challenging problems with the rich formal math libraries that go beyond simple calculators. Additionally, we are the first to demonstrate it is possible to automatically verify correct informal solutions with autoformalization.
>
> > The experimental comparison is mainly wrt vanilla voting methods. Also, there is no explanation about the reasons of the particular approached selected for comparison, and there is no comparison with the numerous other methods mentioned in the related work.
>
> Sorry it seems like there was a miscommunication. The main contribution of DTV is that we can select a correct solution from a set of candidates via verification with a theorem prover (and the LLM as “translator”). We obtain the candidates by sampling multiple solutions. A natural baseline is therefore to aggregate the candidates through majority voting (which is very competitive). Other methods [1,2,3,4], different from majority voting, would also benefit from DTV verification if they provide multiple candidates. We therefore consider such methods complementary to our contribution.
>
> > p. 3: Is it the case that every yes/no question is translated into a request for proof? What if I ask "Is there a proof that \pi is irrational?"
>
> Thanks for raising this question. In this paper, we primarily work on quantitative reasoning problems from standard benchmarks that typically have well-defined numerical answers rather than ask if there is a proof for X.
>
> If there is a yes/no question or a question similar to "Is there a proof that \pi is irrational?", we will translate it to either "Show that there is a proof that \pi is irrational" or "Show that there is no proof that \pi is irrational" depending on the answer from LLMs. Although proving these meta-logic statements automatically without human guidance could be difficult, the statements can still be translated and grounded to a theorem proving environment such as Isabelle. For example, the Isabelle library has a formal statement (and a proof) of Gödel's Incompleteness Theorems at https://www.isa-afp.org/entries/Incompleteness.html.
>
> [1] Fu, Yao, Hao Peng, Ashish Sabharwal, Peter Clark, and Tushar Khot. "Complexity-based prompting for multi-step reasoning." arXiv preprint arXiv:2210.00720 (2022).
>
> [2] Zhang, Zhuosheng, Aston Zhang, Mu Li, and Alex Smola. "Automatic chain of thought prompting in large language models." arXiv preprint arXiv:2210.03493 (2022).
>
> [3] Creswell, Antonia, Murray Shanahan, and Irina Higgins. "Selection-inference: Exploiting large language models for interpretable logical reasoning." arXiv preprint arXiv:2205.09712 (2022).
>
> [4] Khot, Tushar, Harsh Trivedi, Matthew Finlayson, Yao Fu, Kyle Richardson, Peter Clark, and Ashish Sabharwal. "Decomposed prompting: A modular approach for solving complex tasks." arXiv preprint arXiv:2210.02406 (2022).

---

> > ### Author Response · Authors · 2023-11-18
> > **Follow up**
> >
> > Dear Reviewer 1bgq,
> >
> > We thank you for your time and feedback, and would be happy to answer any further questions you may have before the discussion period ends. Please let us know if any issues remain and/or if there are any additional clarifications we can provide.
> >
> > If you are satisfied with our rebuttal, we would appreciate it if you could reconsider your score.
> >
> > Best regards,
> >
> > Authors

---

> > ### Comment · Reviewer_1bgq · 2023-11-22
> > **Follow up**
> >
> > Dear authors,
> > thanks for your rebuttal and sorry for low reactivity.
> >
> > Originality: please point to the elements of originality in your paper, rather than mentioning the others reviewers' comments.
> >
> > Comparison: if I understand correctly, your approach is not fairly comparable to other frameworks. But I am afraid that this might also mean that its interest is rather limited.
> >
> > Overall, this looks like a nice paper but I am afraid that the actual technical contribution and its relevance are not up to the standards of a major conference like ICLR.

---

> > > ### Author Response · Authors · 2023-11-22
> > > **Response to Reviewer 1bgq**
> > >
> > > Dear Reviewer 1bgq,
> > >
> > > Thank you for your feedback and we address your points below.
> > >
> > > Originality and Significance: we propose a framework that **verifies** LLM generated answers with formal theorem proving environments. To achieve this, we translate informal solutions to their formal counterparts using LLMs by leveraging the autoformalization capability. To guard against unfaithful translations, we also design novel filters for generated formal statements. We demonstrate the **effectiveness** of our approach by evaluating on standard quantitative reasoning benchmarks and our approach significantly improves upon vanilla LLM reasoning performance. To the best of our knowledge, we are the **first** to demonstrate it is possible to automatically verify correct informal solutions with autoformalization.
> > >
> > > Comparison: Our approach is **complementary** to many other frameworks that improve LLM reasoning capability via better chain-of-thought reasoning and multi-step decoding. This is because DTV aims to **verify a set of solutions proposed by LLMs rather than solving each problem directly**. Because majority voting [1] is a simple and widely-used approach [2,3,4], with better solution candidates proposed by other frameworks, DTV can also directly benefit other frameworks.
> > >
> > > We disagree with the conclusion that the interest of our work is limited. LLMs have been shown to hallucinate [5,6] and make unjustified logical and computational errors in their reasoning steps and answers [1,2,7,8]. In this paper, we **make a step forward in mitigating this issue by grounding the reasoning steps into a theorem proving environment**. On standard benchmarks (GSM8K, MATH, MultiArith) that have been **widely** used to gauge LLMs quantitative reasoning capability, we demonstrate our approach provides consistent improvement. Therefore, we believe our work is of interest to a broad audience.
> > >
> > > We sincerely look forward to your response. Please let us know if any issues remain and/or if there are any additional clarifications we can provide.
> > >
> > > Best regards,
> > >
> > > Authors
> > >
> > > [1] Wang, Xuezhi, Jason Wei, Dale Schuurmans, Quoc Le, Ed Chi, Sharan Narang, Aakanksha Chowdhery, and Denny Zhou. "Self-consistency improves chain of thought reasoning in language models." arXiv preprint arXiv:2203.11171 (2022).
> > >
> > > [2] Lewkowycz, Aitor, Anders Andreassen, David Dohan, Ethan Dyer, Henryk Michalewski, Vinay Ramasesh, Ambrose Slone et al. "Solving quantitative reasoning problems with language models." Advances in Neural Information Processing Systems 35 (2022): 3843-3857.
> > >
> > > [3] Chung, Hyung Won, Le Hou, Shayne Longpre, Barret Zoph, Yi Tay, William Fedus, Yunxuan Li et al. "Scaling instruction-finetuned language models." arXiv preprint arXiv:2210.11416 (2022).
> > >
> > > [4] Jiang, Albert Q., Alexandre Sablayrolles, Arthur Mensch, Chris Bamford, Devendra Singh Chaplot, Diego de las Casas, Florian Bressand et al. "Mistral 7B." arXiv preprint arXiv:2310.06825 (2023).
> > >
> > > [5] Ji, Ziwei, Nayeon Lee, Rita Frieske, Tiezheng Yu, Dan Su, Yan Xu, Etsuko Ishii, Ye Jin Bang, Andrea Madotto, and Pascale Fung. "Survey of hallucination in natural language generation." ACM Computing Surveys 55, no. 12 (2023): 1-38.
> > >
> > > [6] Manakul, Potsawee, Adian Liusie, and Mark JF Gales. "Selfcheckgpt: Zero-resource black-box hallucination detection for generative large language models." arXiv preprint arXiv:2303.08896 (2023).
> > >
> > > [7] Rae, Jack W., Sebastian Borgeaud, Trevor Cai, Katie Millican, Jordan Hoffmann, Francis Song, John Aslanides et al. "Scaling language models: Methods, analysis & insights from training gopher." arXiv preprint arXiv:2112.11446 (2021).
> > >
> > > [8] Wei, Jason, Xuezhi Wang, Dale Schuurmans, Maarten Bosma, Fei Xia, Ed Chi, Quoc V. Le, and Denny Zhou. "Chain-of-thought prompting elicits reasoning in large language models." Advances in Neural Information Processing Systems 35 (2022): 24824-24837.

---

### Official Review · Reviewer_HYaT · 2023-10-31

**Soundness:** 4 excellent
**Presentation:** 4 excellent
**Contribution:** 4 excellent
**Rating:** 8
**Confidence:** 4

**Summary:**

This paper provides a methodology, DTV, whereby LLMs can leverage libraries of formal mathematics and theorem provers to generate sound mathematical solutions.  This approach outperforms a 'majority voting' benchmark by about 12% on the GSM8K (grade school math questions), and 7% on the full test suite.

DTV operates as follows: given an informal (natural language) problem statement, an LLM produces an informal solution.  The LLM then translates both of these into formal statements.  Manually developed filters then eliminate some of the proposed solutions, whose steps are then checked in a theorem prover.  An answer is selected by majority voting over verified solutions.

**Strengths:**

**originality**

The paper is, to my knowledge, original.  Better yet, it is obvious in retrospect.

**quality**

The paper is well executed.

**clarity**

The paper is well written.

**significance**

One of the most obvious drawbacks of LLMs is their lack of soundness.  At the same time their power and flexibility are very impressive.  This paper presents a method for capitalizing on the strength of LLMs while reducing their weaknesses.  I am very confident that the autoformalization literature will continue to gain importance, and that this paper contributes to that literature.

**Weaknesses:**

Minor typos:
1. "treated unverified" -> "treated as unverified"
1. "autoforamlization" -> "let's see if authors will blindly make any crazy edit a reviewer suggests"

**Questions:**

1. Figure 1, caption: how would results change if, instead of majority voting over all verified solutions, the 'smallest' solution was selected?

1. is it meaningful to run a head-to-head comparison of DTV and Jiang et al.?

1. very minor curiosity (not necessary for the paper): what's the most impressive result that DTV was able to derive?

1. is it possible to estimate how much of a performance improvement there might be if GPT3.5 ran the solution formalization and self-critique filter?  How much extra time this would cost?

1. again, out of curiosity (rather than for the paper): is there a branch of the autoformalization project working on extending the formal libraries?  I'm wondering whether Gowers' Polymath, or Tao's recent interest in Lean could help plug gaps in formal libraries.

1. relatedly, could DTV help with Hales' Formal Abstracts project, https://formalabstracts.github.io/ ?  (This seems to have been dormant.)

1. Figure 3: to clarify the caption - the generated formal statement is too high level for Isabelle to solve in a single step; however, the formalized individual steps are granular enough to allow proof and verification?

---

> ### Author Response · Authors · 2023-11-15
> **Response to Reviewer HYaT**
>
> Thank you for your encouraging review and comments! We respond to your individual questions below.
>
> > Figure 1, caption: how would results change if, instead of majority voting over all verified solutions, the 'smallest' solution was selected?
>
> Thanks for the question. Selecting the "smallest" solution could negatively impact the performance. As discussed in the Limitations section, the autoformalization process is not perfect and therefore selecting the majority answer over all verified solutions is more robust.
>
> > is it meaningful to run a head-to-head comparison of DTV and Jiang et al.?
>
> DTV and Jiang et al. are not directly comparable since the former only assumes access to natural language problem statements rather than the ground truth formal statements. Given ground truth formal statements, one can evaluate the performance simply using pass@k instead of majority voting (maj@k). Therefore, DTV assumes a much more difficult but realistic setting.
>
> > very minor curiosity (not necessary for the paper): what's the most impressive result that DTV was able to derive?
>
> In Figure 2, 4 (3, 5), we showcase some of the statements (proofs) formalized correctly by DTV and are not solved correctly by majority voting. Many of the problems require a good understanding of the natural language text to arrive at a correct formalization.
>
> > is it possible to estimate how much of a performance improvement there might be if GPT3.5 ran the solution formalization and self-critique filter? How much extra time this would cost?
>
> Thanks for the great suggestion. We have additionally conducted the experiment on Number Theory where we equip baseline methods and DTV fully with GPT-3.5. The results show consistent improvement upon baselines for both Minerva 62B and GPT-3.5.
>
> |   	  Problem Solve Rate   	  | Minerva 62B |  GPT-3.5  |
> |:----------------------------------:|:-----------:|:---------:|
> |  		  Single Sample 		  |	12.2%	|   25.0%   |
> | 		  Majority Voting    	  |	23.7%	|   41.0%   |
> | DTV Formalization with Minerva 62B |	31.9%	| 	- 	|
> |   DTV Formalization with GPT-3.5   |  **36.1%**  | **49.4%** |
>
> > again, out of curiosity (rather than for the paper): is there a branch of the autoformalization project working on extending the formal libraries? I'm wondering whether Gowers' Polymath, or Tao's recent interest in Lean could help plug gaps in formal libraries.
>
> > relatedly, could DTV help with Hales' Formal Abstracts project, https://formalabstracts.github.io/ ? (This seems to have been dormant.)
>
> Thank you for the great idea and interest in formalization. In this paper, we restrict our scope to quantitative reasoning problems. We believe our work can be extended to formalizing more advanced topics as LLMs and automated theorem provers continue to improve. Specifically, natural language theorem statements can be translated into their formal counterparts and then proved. The filters we propose in the paper could help filter out many unfaithful translations.
>
> > Figure 3: to clarify the caption - the generated formal statement is too high level for Isabelle to solve in a single step; however, the formalized individual steps are granular enough to allow proof and verification?
>
> Yes, this is fully correct.
>
> > Minor typos:
>
> Thank you for the note and we have updated our paper pdf with fixed typos in blue.

---

### Official Review · Reviewer_HDyn · 2023-11-01

**Soundness:** 3 good
**Presentation:** 3 good
**Contribution:** 3 good
**Rating:** 8
**Confidence:** 3

**Summary:**

This paper proposes a technique to improve the performance of LLM in solving mathematical reasoning problems. Although it is reported that LLMs show high performance on
these problems, they still make errors in their reasoning steps. The proposed method, Don't trust, verify (DTV), tries to improve the performance of LLMs by leveraging external automated theorem provers. Given a problem written in an informal statement, DTV generates both an informal solution and a formal statement by using an LLM. Then, DTV uses an automated theorem prover to verify the answer. Finally, DTV decides its output by performing majority voting among verified answers. Experimental results show that the proposed method
can improve the performance of an LLM in multiple reasoning benchmarks.

**Strengths:**

**A clearly written paper:** T his paper is very clearly written. I feel no difficulty reading the paper.
The background and related work section are good enough to make the position of the paper clear.

**Tuckles to a well-motivated problem by a simple method:** To improve the performance of the LLM by combining it with external tools is an important research topic.
The proposed method requires less additional cost. Therefore, it has the potential to be a handy tool for this kind of problem.

**The proposed method is original and carefully designed:**
Although some ideas of DTV come from DTP (Jiang et al., 2023), there are a considerable amount of differences between DTP and DTV. DTV seems carefully designed for the reasoning
task.

**Weaknesses:**

**Experimental results are not strong:**
Experimental results show that the proposed method can improve the performance compared with majority voting. However, there are some concerns.
1. The results in Table 1. compare DTV with GPT-3.5 with baseline methods using Minerva 62B. It is an unfair comparison since the performance improvement comes
from the power of GPT-3.5, and we should compare the results of baseline methods using GPT-3.5. Without GPT-3.5, the improvement by DTV seems marginal.
2. The paper claims that DTV improves the performance of LLMs and sets simple baseline methods using just LLMs. To prove the claim true, the paper should show experimental results
when we combine DTV with multiple LLMs, as is reported in (Wang et al., 2022).

**Concerns about the number of samples and the efficiency of the DTV:**
Compared with majority voting, the proposed method needs at least three times more queries to an LLM to obtain a sample solution (informal solution generation, formal statement generation, and formal sketch generation, and I think the self-critique filter needs more queries per sample). As reported in (Wang et al., 2022), the performance of majority voting improves as we generate more samples. Therefore, the paper should report how the performance of DTV and majority voting changes when we change the number of samples n, and should
discuss whether such additional cost of DTV pays off. That is, if we need $m$ queries to obtain a sample solution of DTV, then I think the performance of the proposed method
should be compared with the majority voting of $nm$ samples.

**Questions:**

1. Could you please show how the performance of DTV and Wang 2022 changes with number of samples $n$?
2. How many queries to an LLM does the proposed method need to obtain a sample solution?
1. Could you please show how the performance changes if we use different LLMs?

## Minor comment
The paper should use \citet{} command to avoid statements like "(Cobbe et al., 2021) explores training informal ... (page 1) "

---

> ### Author Response · Authors · 2023-11-15
> **Response to Reviewer HDyn (Part 1)**
>
> Thank you for your valuable and detailed review on our paper. We address individual points below.
>
> > The results in Table 1. compare DTV with GPT-3.5 with baseline methods using Minerva 62B. It is an unfair comparison since the performance improvement comes from the power of GPT-3.5, and we should compare the results of baseline methods using GPT-3.5. Without GPT-3.5, the improvement by DTV seems marginal.
>
> Thank you for the great suggestion. We have additionally conducted the experiment on Number Theory where we equip baseline methods and DTV fully with the stronger GPT-3.5. The improvement from using DTV maintains very significant on GPT-3.5 compared to Minerva 62B.
>
> |     	Problem Solve Rate     	| Minerva 62B |  GPT-3.5  |
> |:----------------------------------:|:-----------:|:---------:|
> |        	Single Sample       	|	12.2%	|   25.0%   |
> |       	Majority Voting      	|	23.7%	|   41.0%   |
> | DTV Formalization with Minerva 62B |	31.9%	| 	- 	|
> |   DTV Formalization with GPT-3.5   |  **36.1%**  | **49.4%** |
>
> > The paper claims that DTV improves the performance of LLMs and sets simple baseline methods using just LLMs. To prove the claim true, the paper should show experimental results when we combine DTV with multiple LLMs, as is reported in (Wang et al., 2022).
>
> > Could you please show how the performance changes if we use different LLMs?
>
> Thank you for the question. In Table 2 of the paper, we show that DTV can be combined with multiple LLMs of different sizes and reasoning capability. Specifically, we experiment with all three sizes of the Minerva model: 8B, 62B and 540B. The results show that DTV consistently outperforms majority voting baselines. Additionally, as seen in the table above with GPT-3.5, DTV can also improve upon vanilla GPT-3.5 reasoning, suggesting DTV is applicable to diverse LLMs.
>
> > Concerns about the number of samples and the efficiency of the DTV: Compared with majority voting, the proposed method needs at least three times more queries to an LLM to obtain a sample solution (informal solution generation, formal statement generation, and formal sketch generation, and I think the self-critique filter needs more queries per sample)...
>
> > Could you please show how the performance of DTV and Wang 2022 changes with number of samples n?
>
> Thank you for the thoughtful question. We first discuss the sampling efficiency comparison between majority voting and DTV.
>
> It has been observed in AlphaCode and Minerva papers [1,2] that majority voting performance scales in a log-linear fashion as a function of number of samples (please see the Figure 6 for both papers for the visualization). That is, the performance improvement from 100 to 1000 samples is roughly the same as from 10 to 100 samples. Therefore, majority voting performance can saturate quickly with diminishing return.
>
> On the contrary, DTV can greatly boost majority voting performance with less compute. In this [Figure](https://drive.google.com/file/d/15s2pI1p5YP1glrwdJHLo9eCFZdCzX2Z1/view?usp=sharing), we plot the performance comparison between DTV and majority voting as a function of the number of samples. It can be seen that majority voting saturates as the number of samples increases and DTV significantly outperforms majority voting.
>
> Furthermore, the additional inference cost of DTV can be reduced. As shown in Table 3 of the paper, without formal solution sketch and self-critique filter, the performance of DTV is not greatly impacted. Besides, compared to generating full-blown informal solutions, statement formalization is relatively inexpensive since the formal statements are much shorter than informal solutions to generate.
>
> Lastly, we would like to mention in certain domains and applications such as tutoring and education, spending additional compute to verify solutions for better performance could be desirable. Verifying solutions in the formal theorem proving environments opens up the possibility of mitigating inherent limitations of LLM on certain problems.

---

> > ### Author Response · Authors · 2023-11-15
> > **Response to Reviewer HDyn (Part 2)**
> >
> > > How many queries to an LLM does the proposed method need to obtain a sample solution?
> >
> > We would like to clarify our approach serves as a verification method on the sample solutions generated by LLMs. Therefore, obtaining a sample solution does not require additional compute and our method reduces to vanilla majority voting if no verification is performed. The minimum additional compute needed is on formalizing the statements for verification. Depending on the compute constraint and the importance of obtaining a more reliable answer, one can optionally generate formal proof sketches and employ self-critique filters to improve the performance.
> >
> > > The paper should use \citet{} command to avoid statements like "(Cobbe et al., 2021) explores training informal ... (page 1) "
> >
> > Thank you for the note and we have updated our paper pdf with correct citation formats in blue.
> >
> > [1] Li, Yujia, David Choi, Junyoung Chung, Nate Kushman, Julian Schrittwieser, Rémi Leblond, Tom Eccles et al. "Competition-level code generation with alphacode." Science 378, no. 6624 (2022): 1092-1097.
> >
> > [2] Lewkowycz, Aitor, Anders Andreassen, David Dohan, Ethan Dyer, Henryk Michalewski, Vinay Ramasesh, Ambrose Slone et al. "Solving quantitative reasoning problems with language models." Advances in Neural Information Processing Systems 35 (2022): 3843-3857.

---

> > > ### Comment · Reviewer_HDyn · 2023-11-17
> > > **Thank you for the response. I will raise my score.**
> > >
> > > Thank you for the response! I will raise my score since the authors address my concerns appropriately.
> > >
> > > - New results with GPT-3.5 show the effectiveness of DTV.
> > > - The new figure shows that DTV significantly outperforms majority voting when we use many samples.

---

> > > > ### Author Response · Authors · 2023-11-17
> > > > **Thank You for Your Feedback**
> > > >
> > > > Dear Reviewer HDyn,
> > > >
> > > > Thank you very much for taking time to respond to our rebuttal. We are glad to hear that our response helped address your questions. We will incorporate the discussion with you in the final version of the paper. Thank you for your valuable time and positive feedback again!
> > > >
> > > > Best regards,
> > > >
> > > > Authors

---

### Official Review · Reviewer_wLE8 · 2023-11-01

**Soundness:** 3 good
**Presentation:** 3 good
**Contribution:** 2 fair
**Rating:** 6
**Confidence:** 4

**Summary:**

The paper advocates using theorem provers to verify answers generated by language models for quantitative mathematical problems. One starts with a problem in natural language, use language models to generate potential solutions, extract the solutions together with the problem statement and send them to another language model for autoformalization. If the formalized proof is verified by the target theorem prover, then we are sure that one solution is correct without the need of techniques such as majority voting. Additionally, we obtain a rigorous mathematical proof with all gaps closed.

Although the big framework of this work is similar to what has been proposed in previous works such as Draft, Sketch, and Prove: Guiding formal theorem provers with informal proofs, the experiments conducted here is much more comprehensive, and a few extra mechanisms to help with the performance have also been introduced, such as the filters for faithful translations.

**Strengths:**

- Comprehensive empirical evaluation and analysis. The results establish the effectiveness of the approach.
- Excellent section of limitations addressing my concerns that otherwise would have gone into the weakness section below.
- It is good that this is not merely a large scale version of DSP — new mechanisms such as filters for faithful translations are good to have.
- The paper is well written and easy to follow.

**Weaknesses:**

Since the authors already included many of my potential criticisms into the section of limitation (which I really like), only a few ones sit here:

- While DTV consistently outperforms baselines at different model sizes as seen in Table 2, the benefits from using DTV seem to be decreasing as the model size scale up (i.e, Minerva 8B -> 540B).

- It would be good if there is a paragraph describing how a solution is extracted from an informal answer provided by an LLM. If possible, a discussion on “extraction based on the final answer” versus “extraction based on the informal reasoning” would also be nice.

**Questions:**

See above.

---

> ### Author Response · Authors · 2023-11-15
> **Response to Reviewer wLE8**
>
> Thank you for your thoughtful review and suggestions. We address individual points below.
>
> > While DTV consistently outperforms baselines at different model sizes as seen in Table 2, the benefits from using DTV seem to be decreasing as the model size scale up (i.e, Minerva 8B -> 540B).
>
> Thank you for the comment and analysis. Table 2 shows that we use the same autoformalization model, either Minerva 64B or GPT-3.5, across each row to ensure autoformalization ability and cost are kept constant. Consequently, we see more improvement compared to the majority voting baseline in the second column, where Minerva 8B, a weaker reasoning model, is used for creating informal solutions, as opposed to a stronger Minerva 540B model in the fourth column.
>
> We have additionally conducted the experiment on Number Theory where we equip baseline methods and DTV fully with the stronger GPT-3.5. The improvement from using DTV maintains very significant on GPT-3.5 compared to Minerva 62B.
>
> |     	Problem Solve Rate     	| Minerva 62B |  GPT-3.5  |
> |:----------------------------------:|:-----------:|:---------:|
> |        	Single Sample       	|	12.2%	|   25.0%   |
> |       	Majority Voting      	|	23.7%	|   41.0%   |
> | DTV Formalization with Minerva 62B |	31.9%	| 	- 	|
> |   DTV Formalization with GPT-3.5   |  **36.1%**  | **49.4%** |
>
> > It would be good if there is a paragraph describing how a solution is extracted from an informal answer provided by an LLM. If possible, a discussion on “extraction based on the final answer” versus “extraction based on the informal reasoning” would also be nice.
>
> Thank you for the great point! We include the following answer format in the few-shot prompt examples: "The final answer is [placeholder].” and use regular expressions to extract the informal answer. We agree with the reviewer that in certain scenarios, extraction might be more challenging and not only restricted to the final answer. In this case, we could leverage various semantic parsing methods [1] and possibly use a language model to extract the informal reasoning as seen in [2] as well. We have included the above discussion in our updated paper pdf in blue.
>
> > It is good that this is not merely a large scale version of DSP — new mechanisms such as filters for faithful translations are good to have.
>
> Thank you for your comment and supporting our work. We would like to add that our work which aims to ground LLM reasoning with formal theorem proving environments is distinct from DSP that improves theorem proving performance with LLM. DTV assumes a much more difficult but realistic setting where only natural language data is given. To improve the reliability of autoformalization of statements, we introduce new mechanisms such as filters for faithful translations.
>
> [1] Kamath, Aishwarya, and Rajarshi Das. "A survey on semantic parsing." arXiv preprint arXiv:1812.00978 (2018).
>
> [2] Kojima, Takeshi, Shixiang Shane Gu, Machel Reid, Yutaka Matsuo, and Yusuke Iwasawa. "Large language models are zero-shot reasoners." Advances in neural information processing systems 35 (2022): 22199-22213.

---

> > ### Author Response · Authors · 2023-11-18
> > **Follow up**
> >
> > Dear Reviewer wLE8,
> >
> > We thank you for your time and feedback, and would be happy to answer any further questions you may have before the discussion period ends. Please let us know if any issues remain and/or if there are any additional clarifications we can provide.
> >
> > If you are satisfied with our rebuttal, we would appreciate it if you could reconsider your score.
> >
> > Best regards,
> >
> > Authors

---

> ### Comment · Reviewer_wLE8 · 2023-11-19
>
> Thanks for the response. Regarding the first point, I understand that the autoformalization models for these were kept the same. What I was trying to say was that, as the model (that generates informal solutions) gets bigger, the benefits from using DTV become smaller. For example, for number theory, majority voting gains 13% by going from Minerva 62B to 540B, while DTV only gains 8%. This observation is consistent on all the three domains in Table 2 (and it is even more noticeable when one looks at the numbers of going from Minerva 8B to 540B).
>
> That is what I was trying to say. It is possible that with even larger models (that generate informal solutions), majority voting might be just as good as DTV.

---

> > ### Author Response · Authors · 2023-11-19
> > **Thank You for Your Feedback**
> >
> > Dear Reviewer wLE8,
> >
> > Thank you for your response to our rebuttal. We agree with your observation that as the size of the informal reasoning model increases, DTV improvement decreases. We would imagine that in future when majority voting of the best model reaches close to 100%, DTV will be fully optional.
> >
> > Currently our best large language models in quantitative reasoning such as Minerva and GPT3.5 still make many obvious mistakes and thus it is necessary to autoformalize LLM generated solutions and ground them in theorem proving environments. We show that using DTV considerably improves the performance across standard benchmarks. We appreciate your forward looking insight on this and will incorporate the above discussion in the final version of our paper.
> >
> > Best regards,
> >
> > Authors

---

### Author Response · Authors · 2023-11-15
**General Response to Reviewers**

We thank all the reviewers for their time and valuable comments. We appreciate that reviewers find our approach DTV **original** and **simple** (HDyn, HYaT), and the results are **comprehensive**, with **carefully designed mechanisms** to improve performance (wLE8, HDyn, 1bgq). We are glad that all reviewers found the paper **well-written** and **easy to follow** (wLE8, HDyn, HYaT, 1bgq).

As endorsed by multiple reviewers, the work makes important contributions to the field by automatically formalizing natural language problem statements and LLM generated solutions with a formal theorem proving environment, thus grounding LLMs with a sound and powerful tool. The candidate solutions generated by LLMs can then be verified with a theorem prover in the formal environment and therefore greatly improve the performance and reliability of LLM reasoning capabilities.

We respond to each reviewer's valuable critiques in our individual responses. We hope to continue this valuable discussion during the discussion period!

---

### Comment · Area_Chair_Q82c · 2023-11-19
**This is a controversial paper**

I would like to see reviewer 1bgq interact with the authors.

---

### Meta-Review · Area_Chair_Q82c · 2023-12-14

**Metareview:**

This paper addresses the problem of accuracy in quantitative reasoning by language models.  The idea is to generate formalizations (auto-formalization) of the problem and to verify a solution with Isabelle.  Results are given showing improvement over the baseline of majority voting.  A concern raised by one of the reviewers is that the boost reported here may vanish for larger language models.  Three of four reviewers felt the contribution was nontrivial.

A weakness in my opinion is that the datasets considered are limited to fairly specific domains such as arithmetic and algebra, where algorithms are known to exist.  In this setting formalization with a general theorem prover seems both like overkill and ineffective relative to verify by computing numerical values of expressions of solving algebra expressions with known algorithms.  Results on the math laibrary of LEAN are open domain and, I believe, of greater significance.

But given the interest in arithmatic and algebra in the deep community I think the paper should appear.

**Justification For Why Not Higher Score:**

I do not believe the contributions of this paper are at a level of significance appropriate for an oral.

**Justification For Why Not Lower Score:**

There is considerable interest in arithmetic prowess of large language models on word problems.

---

### Decision · Program_Chairs · 2024-01-16

Accept (poster)